# Interaction-centric Hypersphere Reasoning for Multi-person Video HOI Recognition

## Abstract

Understanding human-object interaction (HOI) in videos represents a fundamental yet intricate challenge in computer vision, requiring perception and reasoning across both spatial and temporal domains, espically in multi-person scenes. HOI encompasses humans, objects, and the interactions that bind them. These three facets exhibit interconnectedness and exert mutual influence upon one another. However, contemporary video HOI recognition methods focus on the utilization of disentangled representations, neglecting their inherent interdependencies. Our key assertions are two-fold: 1) the three HOI facets are inherently interdependent; 2) each HOI possesses a manifold structure in charge of specific interaction class, while human and object entities are both influenced by their respective interaction classes. In light of this, we propose an interaction-centric hypersphere reasoning model for multi-person video HOI recognition. The design of interaction-centric hypersphere visualizes the structure bias of HOI introduced into the model, explicitly directing the learning process towards comprehending the HOI manifold structures governed by interaction classes. Specifically, we design a context fuser to learn the interdependencies among humans, objects and interactions. Furthermore, to equip the model with the capacity for temporal reasoning, we introduce an interaction state reasoner module on top of context fuser. Finally, to depict the manifold structure of HOIs, we adopt an interaction-centric hypersphere and compute the probability of each human-object entity belonging to specific interaction classes. Consequently, our model unravels the intricacies of HOI manifold structure and is flexible for both multi-person and single-person scenarios. Empirical results on multi-person video HOI dataset MPHOI-72 indicate that our method remarkably surpasses state-of-the-art (SOTA) method by more than 22% $F_1$ score. At the same time, on single-person datasets Bimanual Actions (single-human two-hand HOI) and CAD-120 (single-human HOI), our method achieves on par or even better results compared with SOTA methods.

## 1 Introduction

Video-based Human-Object Interaction (HOI) recognition aims to identify the interactions occurring between human and object entities within video frames. Precisely recognizing HOIs in real world scenarios is essential for a bunch of applications, such as assisting patients by recognizing daily activities and predicting pedestrian movements to avoid accidents for autonomous vehicles.

Most existing HOI recognition research focus on static images (Zhang et al., 2022; Liu et al., 2022), with much less attention on video-based HOI recognition. Video-based HOI recognition is more demanding than image-based HOI recognition due to the necessity of comprehending complex spatio-temporal dynamics and reasoning about human and object motions. The complexity is further exacerbated when dealing with multi-person circumstances. In such cases, various human and object entities mutually influence each other, resulting in intricate interdependencies within the scene. Additionally, the three components (*human*, *object* and *interactions* that the entities are involved in) of HOI exhibit interwind structures, *e.g.*, the possible interactions that can occur within a scene given a human and an object. However, current video HOI recognition methods (Qiao et al., 2022; Morais et al., 2021) do not fully explore such inherent structural nature of HOI components. Instead, they often opt for disentangled representations for each components, which may have suboptimal representation capabilities.

To overcome the aforementioned limitations, we introduce an interaction-centric hypersphere approach for representing video HOIs. This approach leverages the concept of a hypersphere, where the *interaction* is located at its center, while the *human* and *object* entities involved in that interaction are situated on the hypersphere's surface. We assume that human-object entities belonging to each interaction class are distributed on their respective hyperspheres, specific to that interaction class. Each hypersphere represents an HOI manifold structure in charge of a specific interaction class. In that sense, we introduce HOI structure bias into the model with the help of hypersphere to visualize that bias, compelling the model to make predictions within the confines of this structure bias. Consequently, we can depict each HOI as a hyperspherical representation characterized by a centroid and radius embedding learned from the model. To enhance the awareness of complex HOI structures in our representations, we introduce the Context Fuser (CF), which encodes both entity representations and interaction representations. Moreover, to empower our model with the ability to reason about interaction state transitions across video frames, we propose the Interaction State Reasoner (ISR) module for generating interaction representations. In addition, we employ a bidirectional Gated Recurrent Unit (BiGRU) to model temporal dynamics across video frames. This multi-level representation learning framework not only facilitates effective exploration of the interdependencies among structured HOI components but also empowers the model with interaction reasoning capabilities in both spatial and temporal domains.

Concretely, we generate context-rich and reasoning-aware video HOI representations through three key components: the Context Fuser (CF), Interaction State Reasoner (ISR) and Bidirectional GRU (BiGRU). The CF module integrates context information from human-object entities and interactions. It comprises three fuser blocks for humans, objects, and interactions. The Object Fuser Block processes local video frame data, enhancing object features. The Interaction Fuser Block combines human and object representations with interaction-specific characteristics. Additionally, in multi-person scenes, the Human Fuser Block captures human representations influenced by others. This approach fosters comprehensive HOI representations via effective context fusion. To facilitate interaction reasoning, we place the ISR module on top of the context fuser module, yielding entity representations capable of capturing interaction transition dynamics. These entity representations are then input into the BiGRU module to model temporal dynamics across video frames, thereby ensuring a comprehensive understanding of the evolving context and interactions within the video data. Finally, we determine interaction classes in each frame with the interaction-centric hypersphere, computing the probability of human-object entities belonging to specific interaction classes.

We assess our model's performance on three video-based HOI datasets: MPHOI-72 (Qiao et al., 2022) (multi-person), Bimanual Actions (Dreher et al., 2020) and CAD-12 (Koppula et al., 2013b) (single-person). Our results highlight our model's superiority in multi-person scenarios, achieving an impressive over $22\%$ $F_1$ score improvement over the current state-of-the-art (SOTA). In single-person scenarios, our method delivers on par or even better performance compared to the current SOTA method. Our major contributions are summarized as follows:

- To represent inherent HOI manifold structures, we propose an interaction-centric hypersphere representation scheme. This scheme explicitly introduce the structure bias of HOI, elucidating the interdenpendencies among its components.

- To learn context-rich and reasoning-aware entity representations, we introduce context fuser and interaction state reasoning modules. This enhancement results in entity representations that are highly suitable for video-based HOI tasks.

- Extensive experiment results showcase that our method achieves SOTA performance with a huge improvement of more than $22\%$ $F_1$ score over existing methods in multi-person scenario. Additionally, our model achieves competitive results in single-person cases compared to SOTA method.

## 2 RELATED WORKS

**HOI detection in images**: HOI detection in images aims at understanding interactions in images between humans and objects. Different methods have been proposed in previous studies. Some works propose Convolutional Neural Networks (CNN)-based methods which can be further divided into one-stage methods (Liao et al., 2020; Zhong et al., 2021; Kim et al., 2020) and two-stage methods (Li et al., 2019; Gao et al., 2020; Wang et al., 2019; Gupta et al., 2019). However, these

methods usually lack of ability to capture global context information. Recently, Transformer-based models (Kim et al., 2021; Tamura et al., 2021; Zhang et al., 2022; Iftekhar et al., 2022) became the main approach for the HOI task. Following the architecture of DETR, these models achieved superior performance on HOI detection. Moreover, some works also utilize other methods including graph (Park et al., 2023), interactiveness field (Liu et al., 2022) and compositional prompt tuning Gao et al. (2022) to get better performance. These various approaches to image HOI provide the fundamentals for video HOI recognition.

**HOI recognition in videos**: Video-based HOI recognitions have to deal with both spatial and temporal reasoning. Before the use of neural networks, some early studies formulated this task using the Markov model (Koppula et al., 2013b) to utilize temporal cues. In Nagarajan et al. (2019), HOI hotspots in videos are learned in a novel approach, with two networks trained jointly to capture spatial regions where actions happen. Recent works have used Recurrent Neural Networks (RNN) combined with Graph Neural Networks (GNN) (Qi et al., 2018; Qiao et al., 2022; Morais et al., 2021; Sunkesula et al., 2020) to predict human-object relations in videos. Inspired by ViT (Dosovitskiy et al., 2020), some works also propose Transformer-based methods to reason spatial relations better (Tu et al., 2022). However, RNN-based models usually require complex training strategies or long training time in order to achieve the best performance. Moreover, when multiple persons are involved in an activity jointly, these methods lack the ability to model their collaboration, resulting in poor performance when the interactions are performed by multiple persons.

**Hyperspheres for class representation**: Hyperspheres have been demonstrated to be an effective approach to model class representation (Mettes et al., 2019; Deng et al., 2022). Geometrical modeling strategy has been proposed in Deng et al. (2022), where the effectiveness has been confirmed. This approach proves advantageous for capturing and representing enriched class-level information, particularly well-suited for creating measurements in Euclidean space. Consequently, it is naturally adaptable to structured prediction tasks.

## 3 MOTIVATION

Video HOI recognition task involves identifying both the human and object entities engaged in an interaction across a sequence of video frames. This task encompasses spatial and temporal aspects, as it requires understanding the relationships between humans, objects, and their interactions over time. However, current methods for video HOI recognition often neglect this crucial dependency structure, resulting in the separation of learned representations associated with humans, objects, and their interactions, ultimately compromising their representational accuracy. To address this issue, we propose a novel approach, introducing HOI structure bias into our model and visualize the bias with interaction-centric hypersphere. Additionally, we introduce a context fuser and an interaction state reasoner in our model to facilitate the learning of context-rich and reasoning-aware entity representations.

## 4 METHOD

### 4.1 PROBLEM FORMULATION

For a video dataset $\mathcal{V}$, given a video clip $V \in \mathcal{V}$ containing $T$ video frames $\{v_1, ..., v_T\}$, video HOI recognition aims to predict the temporal segmentation of interactions between human and object entities across all the video frames. Formally, we aim to learn an HOI recognition model $\mathcal{M}$ that outputs the segmentation of human's sub-activity $\{s_n\}_{n=1}^{N}$ in each frame, where $N$ is the number of human sub-activity segments. Each segment $s_n$ is represented as an interval from its start time $t_n$ to end time $t_{n+1}$: $s_n = [t_n, t_{n+1})$. The start and end time of each segment are determined from interaction probability prediction $\{u^t\}_{t=1}^{T}$ of each frame, where $u^t \in \mathbb{R}^K$, $K$ is the number of possible interaction classes.

### 4.2 MODEL DESIGN

In the following section, we introduce our interaction-centric hypersphere reasoning model for video HOI recognition in detail. As shown in Fig. 1, our major idea is to construct a hypersphere to represent each HOI in the scene. For each hypersphere, the *interaction* locates at the center of the hypersphsere, while the corresponding *human-object* entity belongs to that *interaction* locates at the surface of the hypersphere. We construct Context Fuser (CF) module to learn context-rich human-

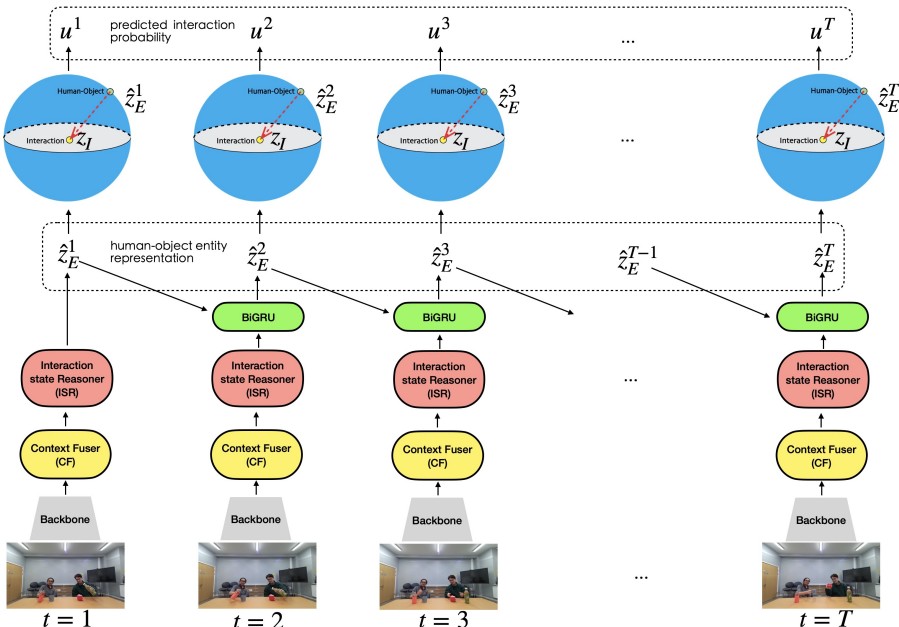

Figure 1: **Model Overview**. Each video frame is applied with a backbone for feature extraction. Subsequently, a context fuser and interaction state reasoner is employed for learning interaction representations $z_I$ and human-object entity representations $\{\hat{z}_E^t\}_{t=1}^T$. Bidirectional GRU is further utilized to model temporal dynamics across video frames. The final predicted interaction probability $\{u^t\}_{t=1}^T$ is computed from the interaction-centric hypersphere.

object entity representations. For the aim of enabling model with reasoning ability over interaction state transitions, we propose Interaction State Reasoner (ISR) to reason on whether the current interaction will be continued or stopped. To model the temporal dynamics of HOI in videos, we update human-object entity representations $\{\hat{z}_E^t\}_{t=1}^T$ along the temporal domain with bidirectional GRU (BiGRU). Finally, predicted interaction class probability $u^t$ of each frame is computed from the interaction-centric hypersphere.

### 4.2.1 CONTEXT FUSER

For a sequence of video frames $\{v_t\}_{t=1}^T$, we follow 2G-GCN (Qiao et al., 2022) to extract feature of humans and objects from backbone network. The extracted human features $z_H \in \mathbb{R}^d$ contain both bounding box information and skeleton keypoint information, where $d$ indicates the feature dimension. Object features $z_O \in \mathbb{R}^d$ contain only bounding box information.

We design a context fuser (CF) module shown in Fig. 2 to generate human-object entity representations $\{z_E^{\bar{t}}\}_{t=1}^T$ based on human, object and contextual information. In multi-person circumstances, CF contains object fuser block, interaction fuser block and human fuser block sequentially. First, we design an object fuser block to incorporate object representations in local temporal regions into current object representations, generating learned object representation $\hat{z}_{O_1}$ and $\hat{z}_{O_2}$:

$$\hat{z}_{O_1} = FFN(SA(Q = z_{O_1}^t; K, V = z_{O_1}^{\bar{t}}) + z_{O_1}^t), \ \hat{z}_{O_2} = FFN(SA(Q = z_{O_2}^t; K, V = z_{O_2}^{\bar{t}}) + z_{O_2}^t), \tag{1}$$

where $SA$ indicates self-attention, $FFN$ is feed forward network, $z_{O_1}^t \in \mathbb{R}^d$ and $z_{O_2}^t \in \mathbb{R}^d$ are the initial object features, while $z_{O_1}^{\bar{t}} \in \mathbb{R}^{20d}$ and $z_{O_2}^{\bar{t}} \in \mathbb{R}^{20d}$ are stacked object features from a local time window, $\bar{t} \in [t-10, t+10)$. Subsequently, for all the $K$ possible interactions $\{I_i\}_{i=1}^K$ as shown in Fig. 2, each interaction class $I_i$ is prompted as a sentence $s =$"The human is [interact]ing in the scene.", where [interact] indicates the specific interaction class. Then the sentence is applied with the text encoder ($\mathcal{F}_T$) of large-scale vision-language model CLIP (Radford et al., 2021) to initialize the interaction feature $z_I = \mathcal{F}_T(s) \in \mathbb{R}^d$. We also generate context feature $z_C \in \mathbb{R}^d$ to represent the semantic information of each frame. Specifically, we extract frame caption $c_i$ from video frame $i (i = 1, ..., T)$ with BLIP (Li et al., 2022) model and apply the caption with CLIP model to extract text embedding. In order to adapt the human ($z_{H_1}, z_{H_2}$) features, object ($z_{O_1}, z_{O_2}$)

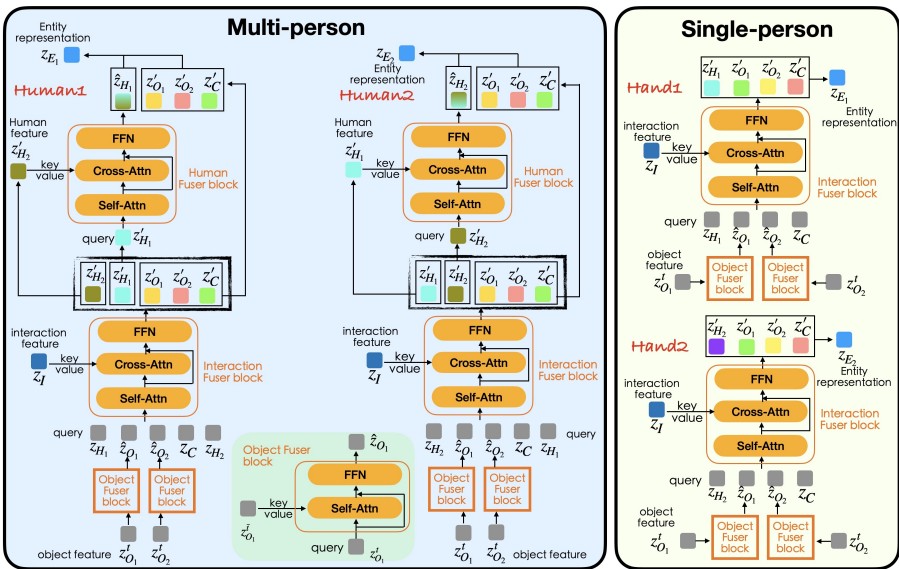

Figure 2: **Context fuser**. In multi-person scenario (left), the context fuser consists of object fuser block, interaction fuser block and human fuser block. In single-person case (right), only object fuser block and interaction fuser block are employed.

features and context feature ($z_C$) to the specific interaction feature $z_I$, we construct an interaction fuser block that contains a cross-attention (CA) module followed by a Feed Forward Network (FFN), generating interaction-aware human features ($z'_{H_1} \in \mathbb{R}^d$, $z'_{H_2} \in \mathbb{R}^d$), object features ($z'_{O_1} \in \mathbb{R}^d$, $z'_{O_2} \in \mathbb{R}^d$) and context feature ($z'_C \in \mathbb{R}^d$). Furthermore, to model the influence between the two humans in multi-person scenarios, we construct a human fuser block, featuring the same architecture with interaction fuser block. For the first human (Human1), the updated human feature $\hat{z}_{H_1} \in \mathbb{R}^d$ is generated as:

$$\hat{z}_{H_1} = FFN(CA(Q = SA(z'_{H_1}); K, V = z'_{H_2}) + SA(z'_{H_1})). \tag{2}$$

Human2 feature $\hat{z}_{H_2} \in \mathbb{R}^d$ is generated in the same way as Eq. 2. Finally, the human-object entity representation $z_{E_1} \in \mathbb{R}^d$ of Human1 is computed by max-pooling operation over all the $d$ dimensions of the four representations shown in Eq. 3:

$$z_{E_1} = MaxPool(\hat{z}_{H_1}, z'_{O_1}, z'_{O_2}, z'_C). \tag{3}$$

The human-object entity representation $z_{E_2} \in \mathbb{R}^d$ of Human2 is computed with similar approach as Eq. 3. The CF module for single-person cases are similar with multi-person, except that the human fuser block is removed and there is only one human feature as query to be fed into the interaction fuser block.

### 4.2.2 INTERACTION STATE REASONER

To augment the model's ability in interaction state transition reasoning, we introduce an Interaction State Reasoner (ISR) module following CF module. ISR module explicitly empowers the model to determine whether the current interaction should persist or transit to another interaction. Specifically, as shown in Fig. 3, at each time $t$, the two possible states $state_1$ and $state_2$ represent "continue" or "stop" of an interaction, respectively. Each state is prompted as one sentence, where $s_1$ ="This interaction is going to continue." and $s_2$="This interaction is going to stop and change to another interaction.". Then the embeddings of the two states $z_{state_1}$ and $z_{state_2}$ are generated from CLIP (Radford et al., 2021) text encoder $\mathcal{F}_T$: $z_{state_1} = \mathcal{F}_T(s_1) \in \mathbb{R}^d$, $z_{state_2} = \mathcal{F}_T(s_2) \in \mathbb{R}^d$. Interaction state embeddings $z_{state_1}$ and $z_{state_2}$ are further fed to a reasoner block (shown in Fig. 3) together with the interaction embedding $z_{\hat{I}}^t \in \mathbb{R}^d$ at time $t$, generating state-informed interaction embeddings $\hat{z}_{state_1}$ and $\hat{z}_{state_2}$. The reasoner block contains a FFN and a State Interpolation (SI) module. The SI module generates the weights $\omega_1, \omega_2$ for the two interaction states in the following approach:

$$\omega_1, \omega_2 = Softmax(FFN(z_E^t) \cdot [z_{state_1}^\top, z_{state_2}^\top]), \tag{4}$$

where $\top$ indicates transpose operation. Subsequently, the final human-object entity representation $\hat{z}_E^t \in \mathbb{R}^d$ at time $t$ is generated by interpolate over the current entity representation $z_E^t$ at time $t$ and the entity representation $z_E^{t-1}$ at time $t-1$:

$$\hat{z}_E^t = \omega_1 \cdot z_E^{t-1} + \omega_2 \cdot z_E^t. \tag{5}$$

Consequently, the generated human-object entity representation $\hat{z}_E^t$ is able to reason on the possible future interaction state transitions.

### 4.2.3 INTERACTION-CENTRIC HYPERSPHERE

With the above generated human-object entity representation $\hat{z}_E^t$ and interaction representation $z_I$, we need to calculate the probability of a human-object entity $E$ categorizing into the interaction class $I_i$. To that end, we design an interaction-centric hypersphere with interaction at the center of hypersphere and human-object entity at its surface. This hypersphere design models the manifold structure of HOI, which is in charge of the specific interaction class. Concretely, we employ a hyperspherical measurement:

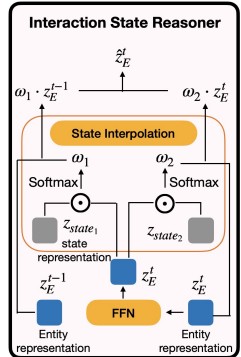

Interaction State Reasoner. $\odot$ indicates inner product.

$$\mathcal{U}(E, I_i) = \frac{\exp(-[||z_{I_i} - \hat{z}_E||_2 - \lambda]_+)}{\Sigma_{j=1}^K \exp(-[||z_{I_j} - \hat{z}_E||_2 - \lambda]_+)}, \tag{6}$$

where $z_{I_i}$ and $z_{I_j}$ indicate the interaction representations of interaction class $i$ and $j$, respectively. $\hat{z}_E$ denotes the human-object entity representation. $[s]_+ \triangleq max(0, s)$. $\lambda$ indicates the radius of the hypersphere, which is set to be a constant in our model. The higher value of $\mathcal{U}(E, I_i)$ suggests that the human and object entity $E$ is more likely to be categorized into $I_i$.

### 4.3 LEARNING OBJECTIVE

The learning objective of our model contains two parts: (i) focal loss $\mathcal{L}_{cls}$ for interaction classification; (ii) interaction feature loss $\mathcal{L}_{fea}$ that controls the smoothness of interaction features in local region.

**Focal loss $\mathcal{L}_{cls}$:** We employ focal loss (Lin et al., 2017) for interaction classification, mitigating the interaction class imbalance problem on model performance. For each video frame $v_i(i = 1, ..., T)$, our model predicts the probability $\hat{y}_i \in \mathbb{R}^K$ of all the interaction classes. The corresponding ground-truth of interaction class $y_i \in \mathbb{R}^K$ is a binary vector. For each intraction class $k$, the focal loss $\mathcal{L}_{cls}^k$ is formulated as: $\mathcal{L}_{cls}^k = -(1 - p_k)^\gamma \log(p_k)$, where $\gamma$ is a hyperparameter to control the focusing extent, $p_k$ is defined as: $\{p_k = \hat{y}_i^k, \text{ if } y_i^k = 1; p_k = 1 - \hat{y}_i^k, \text{ otherwise}\}$. Subsequently, the focal loss $\mathcal{L}_{cls}$ of each video frame is obtained by combining the focal loss of each individual interaction class $k$: $\mathcal{L}_{cls} = \Sigma_{k=1}^K \mathcal{L}_{cls}^k$.

**Interaction feature loss $\mathcal{L}_{fea}$:** We introduce interaction feature loss $\mathcal{L}_{fea}$ to control the temporal smoothness of interaction features. Our model outputs the feature of each human-object entity $E$ in each frame. Inspired by Chopra et al. (2005), in order to improve the continuity, we minimize the feature distance in the same segment and maximize the distance between different segments for each subject. Denote $u_E^t$ as whether the interaction will continue or change to another action for entity $E$ at time $t$. $u_E^t = 1$ indicates the interaction will stop and change to another action at time $t$ for entity $E$ and $u_E^t = 0$ otherwise. We minimize

$$\mathcal{L}_{fea} = \frac{1}{2}\Sigma_{t=0}^{T-1}[(1 - u_E^t)(||z_E^t - z_E^{t+1}||_2)^2 + u_E^t(max(L - ||z_E^t - z_E^{t+1}||_2), 0)^2], \tag{7}$$

where $L$ is a threshold that controls the minimal feature distance when interaction will change. In total, the overall loss is written as:

$$\mathcal{L} = \mathcal{L}_{cls} + \alpha\mathcal{L}_{fea}, \tag{8}$$

where $\alpha$ is a hyperparameter to control the weight of each loss.

### 4.3.1 MODEL INFERENCE

During model inference, we compute the interaction probability $\hat{y}_i \in \mathbb{R}^K$ for each video frame. The interaction class $a$ with the highest probability is chosen as the predicted interaction for that frame: $a = \arg\max_i \hat{y}_i$.

## 5 EXPERIMENTS

### 5.1 EXPERIMENTAL SETUP

**Datasets:** We evaluate our method on MPHOI-72, Bimanual Actions, and CAD-120 datasets, each representing multi-person collaboration, one person with two hands and a single hand respectively.
(I) MPHOI-72 dataset is proposed in Qiao et al. (2022), which consists of multiple humans and objects in the scene. The dataset comprises 72 videos featuring 3 human subjects and 6 objects. Within each video, 2 individuals are paired to engage in 3 distinct activities, encompassing a total of 13 sub-activities, while utilizing 2 to 4 objects
(II) Bimanual Actions dataset (Dreher et al., 2020) is the first HOI dataset to include two hands for subjects to perform interactions which is common in reality. There are 540 videos with one person performing activities with both hands. There are 6 subjects performing 9 different activities with 10 repetitions. There are a total of 14 action labels assigned to each hand, and entity-level annotations are provided on a per-frame basis within the video.
(III) CAD-120 dataset (Koppula et al., 2013a) is popular for HOI recognition. It contains 120 videos with 10 activities performed by 4 participants. There are 10 human sub-activities labeled per frame.
**Evalutaion Metric:** We report $F_1@k$ metric (Lea et al., 2017) with thresholds $k$ = 10%, 25%, and 50%. Compared to frame-based metrics which evaluate prediction on every single frame, this metric could measure prediction continuity in action segments because it views each predicted action segment as correct only when it has the Interaction over Union (IoU) with the corresponding ground truth over the threshold $k$.

### 5.2 IMPLEMENTATION DETAILS

In the experiment, we use three layers of context fuser for Bimanual and two for CAD-120 and MPHOI. The features of humans and objects are extracted from Qiao et al. (2022) and their dimension is mapped to 768, 256, and 512 for MPHOI, Bimanual, and CAD-120 respectively. More details can be found in appendix.

### 5.3 QUANTITATIVE RESULTS

**Multi-person HOI recognition** The quantitative results of joined segmentation and label recognition of sub-activity on MPHOI-72 in Tab. 1 show the performance of our method in multi-person HOI circumstance. Our method outperforms SOTA method 2G-GCN (Qiao et al., 2022) by a large margin in all the three evaluation metrics. For $F_1@10$, $F_1@25$ and $F_1@50$ scores, our method surpasses 2G-GCN 23.0%, 23.7% and 22.5%, respectively. The significant improvement achieved by our method indicates that the human fuser block in the CF module effectively improves the context-aware human representation learning under multi-person scenarios.

**Single-person HOI recognition** We show the quantitative results for single-person HOI recognition in Tab. 2 and Tab. 3, which are performed on CAD-120 and Bimanual Actions datasets, respectivey. Results in Tab. 2 show that our method performs slightly better on CAD-120 dataset compared to SOTA method 2G-GCN, with around 1% improvement over 2G-GCN on all the three metrics. For the Bimanual Actions dataset, our method performs as good as 2G-GCN in $F_1@10$ while achieves 0.9% and 5.0% higher than 2G-GCN in $F_1@25$ and $F_1@50$ score, respectively. These results indicate that our method achieves generally on par or even better performance on single-person video HOI recognition task.

### 5.4 ABLATION STUDY

In this section , we ablate the CF module and the ISR module for validating the effectiveness of these proposed components. As shown in Tab. 1, removing CF module results in more than 20% $F_1$ score drop of the three metrics in MPHOI-72 dataset, indicating the essential improvement of CF module in learning context-rich representations. Visualization results of temporal segmentation of interactions in Fig. 5 indicates that removing CF module results in incorrect interaction predictions in both humans (highlighted in red in Fig. 5). Similarly, in CAD-120 and Bimanual Actions datsets, deleting CF module also results in massive $F_1$ score drop. The visualization results in Fig. 6 suggests an incorrect prediction segment when removing CF module.

Furthermore, we ablate hypersphere by replacing it with Euclidean distance, where $\lambda = 0$ in Eq. 6. Results in Tab. 1, 2 and 3 indicate that utilizing Euclidean distance results in at least 7%, 10% and 3% $F_1$ scores drop in MPHOI, CAD-120 and Bimanual Actions dataset, respectively. Therefore,

Table 1: The results of joined segmentation and label recognition of sub-activity on MPHOI-72. $\Delta$CF: removing context fuser module; $\Delta$ISR: removing interaction state reasoner module; $\Delta\mathcal{L}_{fea}$: removing interaction feature loss; $\lambda = 0$: employing Euclidean distance; $\Delta$CLIP + BLIP: removing CLIP and BLIP models. The improvements of our method over current SOTA method is highlighted with upward arrows.

| Model | $F_1$@10 | $F_1$@25 | $F_1$@50 |
|---|---|---|---|
| ASSIGN (Morais et al., 2021) | $59.1 \pm 12.1$ | $51.0 \pm 16.7$ | $33.2 \pm 14.0$ |
| 2G-GCN (Qiao et al., 2022) | $68.6 \pm 10.4$ | $60.8 \pm 10.3$ | $45.2 \pm 6.5$ |
| **Ours** | $\mathbf{91.6 \pm 0.9(\uparrow 23.0)}$ | $\mathbf{84.5 \pm 2.6(\uparrow 23.7)}$ | $\mathbf{67.7 \pm 2.2(\uparrow 22.5)}$ |
| **Ours** ($\Delta$CF) | $65.8 \pm 12.4$ | $57.6 \pm 14.0$ | $39.2 \pm 12.6$ |
| **Ours** ($\Delta$ISR) | $80.1 \pm 5.5$ | $73.0 \pm 8.2$ | $55.6 \pm 6.1$ |
| **Ours** ($\Delta\mathcal{L}_{fea}$) | $73.5 \pm 15.7$ | $69.7 \pm 13.3$ | $48.8 \pm 13.0$ |
| **Ours** ($\lambda = 0$) | $81.2 \pm 0.7$ | $74.8 \pm 4.2$ | $53.2 \pm 0.3$ |
| **Ours** ($\Delta$CLIP+BLIP) | $80.0 \pm 6.7$ | $73.0 \pm 9.9$ | $55.6 \pm 7.4$ |
| **Ours (Traditional Classifier)** | $84.6 \pm 7.2$ | $74.7 \pm 10.5$ | $54.6 \pm 13.7$ |

Table 2: The results of joined segmentation and label recognition of sub-activity on CAD-120. The notations are the same with Tab. 1.

| Model | $F_1$@10 | $F_1$@25 | $F_1$@50 |
|---|---|---|---|
| rCRF (Sener & Saxena, 2015) | $65.6 \pm 3.2$ | $61.5 \pm 4.1$ | $47.1 \pm 4.3$ |
| Independent BiRNN (Qiao et al., 2022) | $70.2 \pm 5.5$ | $64.1 \pm 5.3$ | $48.9 \pm 6.8$ |
| ATCRF (Koppula & Saxena, 2015) | $72.0 \pm 2.8$ | $68.9 \pm 3.6$ | $53.5 \pm 4.3$ |
| Relational BiRNN (Qiao et al., 2022) | $79.2 \pm 2.5$ | $75.2 \pm 3.5$ | $62.5 \pm 5.5$ |
| ASSIGN (Morais et al., 2021) | $88.0 \pm 1.8$ | $84.8 \pm 3.0$ | $73.8 \pm 5.8$ |
| 2G-GCN (Qiao et al., 2022) | $89.5 \pm 1.6$ | $87.1 \pm 1.8$ | $76.2 \pm 2.8$ |
| **Ours** | $\mathbf{90.7 \pm 2.9(\uparrow 1.2)}$ | $\mathbf{88.1 \pm 2.8(\uparrow 1.0)}$ | $\mathbf{77.6 \pm 4.7(\uparrow 1.4)}$ |
| **Ours** ($\Delta$CF) | $81.1 \pm 4.0$ | $77.0 \pm 4.8$ | $65.2 \pm 5.6$ |
| **Ours** ($\Delta$ISR) | $88.5 \pm 3.7$ | $85.5 \pm 3.6$ | $73.9 \pm 5.7$ |
| **Ours** ($\Delta\mathcal{L}_{fea}$) | $89.3 \pm 1.9$ | $85.6 \pm 2.1$ | $75.9 \pm 4.4$ |
| **Ours** ($\lambda = 0$) | $72.0 \pm 4.4$ | $65.0 \pm 6.9$ | $48.6 \pm 6.3$ |
| **Ours** ($\Delta$CLIP+BLIP) | $89.4 \pm 2.3$ | $85.5 \pm 3.9$ | $74.9 \pm 5.7$ |
| **Ours (Traditional Classifier)** | $79.5 \pm 11.0$ | $73.9 \pm 11.4$ | $56.6 \pm 12.5$ |

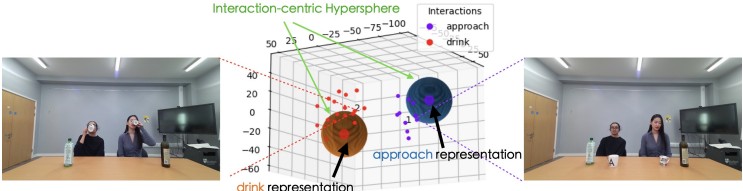

Figure 4: Visualization of interaction-centric hyperspheres, the learned interaction representations (large points locate at the center of hyperspheres) and human-object entity representations (small points surrounding the hyperspheres) in embedding space. Selected frame samples are shown for each interaction-centric hypersphere.

we conclude that Euclidean distance do not introduce HOI structure priors, ignoring valuable structure information of HOI for guiding predictions. Subsequently, we ablate CLIP and BLIP models by randomly initialize interaction features and context features. Results in Tab. 1, 2 and 3 indicate that removing CLIP and BLIP models results in some drop of model performance, but is still on par with or better than SOTA methods. Thus, it is the intricately designed structure of our model that substantiates the substantial enhancement in performance. Finally, we ablate the interaciton-centric hypersphere by replacing it with a traditional classifier constructed with multi-layer perceptron (MLP). The outcomes, as presented in Tab. 1, 2 and 3, reveal a notable decline of over $7\%$ in the $F_1$ score within the MPHOI dataset when employing the traditional classifier. Likewise, in the CAD-120 dataset, the traditional classifier results in a substantial decrease of more than $11\%$ in the $F_1$ score. Additionally, within the Bimanual Actions dataset, the traditional classifier induces a decline exceeding $2\%$ in the $F_1$ score. These findings unanimously underscore the efficacy of the structural bias introduced by the hypersphere module.

## 5.5 QUALITATIVE RESULTS

We show some visualization results on MPHOI in Fig 7 to compare our method with SOTA method 2G-GCN. The red highlighted boxes indicate that 2G-GCN tend to generate unreasonable interaction

Table 3: The results of joined segmentation and label recognition of sub-activity on Bimanual Actions. The notations are the same with Tab. 1.

| Model | $F_1$@10 | $F_1$@25 | $F_1$@50 |
|---|---|---|---|
| Dreher *et al.* (Dreher et al., 2020) | $40.6 \pm 7.2$ | $34.8 \pm 7.1$ | $22.2 \pm 5.7$ |
| Independet BiRNN (Qiao et al., 2022) | $74.7 \pm 7.0$ | $72.0 \pm 7.0$ | $61.8 \pm 7.3$ |
| Relational BiRNN (Qiao et al., 2022) | $77.7 \pm 3.9$ | $75.0 \pm 4.2$ | $64.8 \pm 5.3$ |
| ASSIGN (Morais et al., 2021) | $84.0 \pm 2.0$ | $81.2 \pm 2.0$ | $68.5 \pm 3.3$ |
| 2G-GCN (Qiao et al., 2022) | $\mathbf{85.0 \pm 2.2}$ | $82.0 \pm 2.6$ | $69.2 \pm 3.1$ |
| **Ours** | $85.0 \pm 2.5$ | $\mathbf{82.9 \pm 2.9(\uparrow 0.9)}$ | $\mathbf{74.2 \pm 4.3(\uparrow 5.0)}$ |
| **Ours** ($\Delta$CF) | $82.5 \pm 5.0$ | $80.5 \pm 5.5$ | $71.1 \pm 7.0$ |
| **Ours** ($\Delta$ISR) | $84.1 \pm 2.3$ | $81.8 \pm 2.8$ | $73.0 \pm 3.7$ |
| **Ours** ($\Delta\mathcal{L}_{fea}$) | $84.5 \pm 4.6$ | $82.0 \pm 5.2$ | $71.8 \pm 6.9$ |
| **Ours** ($\lambda = 0$) | $76.7 \pm 5.2$ | $74.3 \pm 6.0$ | $65.2 \pm 6.3$ |
| **Ours** ($\Delta$CLIP+BLIP) | $84.3 \pm 1.4$ | $81.8 \pm 1.8$ | $73.2 \pm 2.7$ |
| **Ours (Traditional Classifier)** | $82.0 \pm 3.6$ | $79.8 \pm 4.1$ | $71.0 \pm 5.6$ |

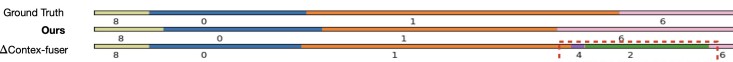

Figure 5: **Qualitative ablation study results on MPHOI-72 dataset**. Major prediction errors are highlighted in red dashed boxes.

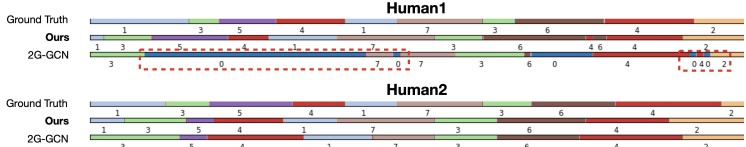

Figure 6: **Qualitative ablation study results on CAD-120 dataset**. Major prediction errors are highlighted in red dashed boxes.

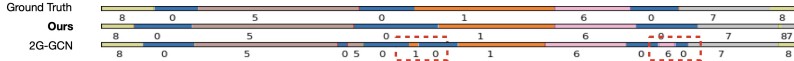

Figure 7: **Visualization results on MPHOI-72 dataset**. Major prediction errors are highlighted in red dashed boxes.

Figure 8: **Visualization results on CAD-120 dataset**. Major prediction errors are highlighted in red dashed boxes.

predictions, while our method generates more reasonable interaction predictions. The visualization results in Fig. 8 show similar prediction pattern where 2G-GCN predicted some unreasonable short segments (highlighted in red boxes) while our method predicts more accurately. We also visualize the interaction-centric hypersphere, the learned interaction representations and the human-object entity representations in embedding space in Fig. 4. Results in Fig. 4 show that human-object entity representations (small dots in Fig. 4) belonging to the specific interaction class locates near the surface of the corresponding hypersphere. These results suggest that our model successfully model the manifold structure of HOI.

## 6 CONCLUSION

In this work, we propose an interaction-centric hypersphere reasoning network for multi-person video HOI recognition. Specifically, we represent HOI components with an interaction-centric hypersphere for class representation. We further propose a context fuser and an interaction state reasoner to learn context-rich and reasoning-aware entity representation. Experiment results show that our method outperforms SOTA method by more than $22\%$ $F_1$ score in multi-person scenarios, and achieves competitive results on single-person cases compared to SOTA methods.

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
