# APPENDIX

In this appendix, we will show

- The impact of hypersphere radius $\lambda$ on model performance.
- More visualization of prediction results.
- Source code of our model.

## 1 MORE IMPLEMENTATION DETAILS

We optimize model parameters using ADAM optimizer (Kingma & Ba, 2014) on 4 NVIDIA TITAN Xp GPUs with a learning rate of $10^{-4}$ and decay the learning rate by half for every 50 epochs. The videos are downsampled by every three frames for training. For model selection, we use 10% of the training data for validation. The loss weight $\alpha$ in Eq.8 is set to be $0.1$. The hypersphere radius $\lambda$ in Eq.6 is set to be 25, 23 and 20 for MPHOI-72, CAD-120 and Bimanual Actions datasets, respectively. The effects of $\lambda$ on model performance are shown in Supplement materials. We employ the same dataset split scheme with 2G-GCN. For MPHOI-72 dataset, we leave two subjects as test set and utilize the remaining subjects for training. For Bimanual Actions and CAD-120 datasets, leave-one-subject cross-validation scheme is employed, evaluating the generalization ability of our method.

## 2 THE SENSITIVITY OF HYPERSPHERE RADIUS $\lambda$

In this section, we show the influence of hypersphere radius $\lambda$ on the model performance. For each dataset, we choose five $\lambda$ values to validate model performance under each specific value. In MPHOI-72 and CAD-120 datasets, $\lambda = \{15, 23, 25, 27, 40\}$. In Bimanual Actions dataset, $\lambda = \{5, 10, 15, 20, 25\}$. The results in Fig. 1, Fig. 2 and Fig. 3 show that our model achieves best performance in MPHOI-72 dataset with $\lambda = 25$, CAD-120 dataset with $\lambda = 23$ and Bimanual Actions with $\lambda = 20$.

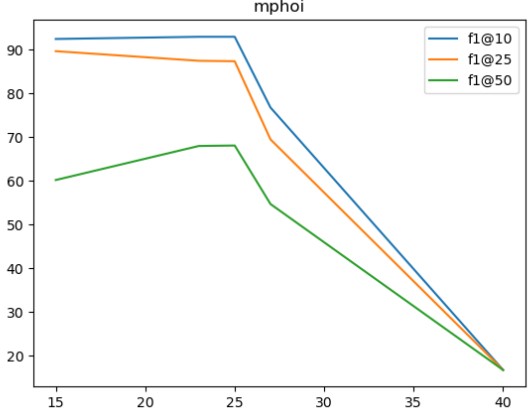

Figure 1: The influence of different hypersphere radius $\lambda$ on model performance in MPHOI-72 dataset.

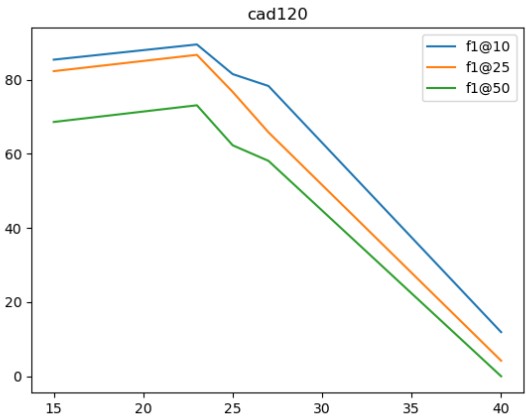

Figure 2: The influence of different hypersphere radius $\lambda$ on model performance in CAD-120 dataset.

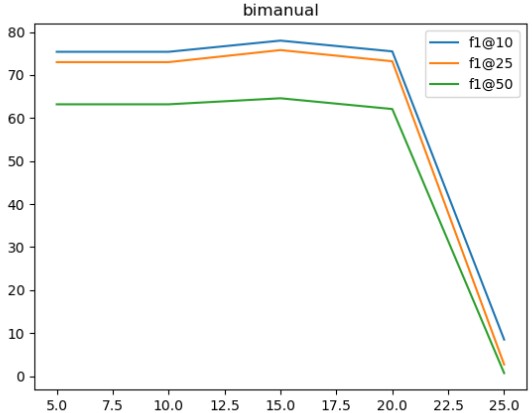

Figure 3: The influence of different hypersphere radius $\lambda$ on model performance in Bimanual Actions dataset.

## 3   MORE VISUALIZATION OF PREDICTION RESULTS

In this part, we show more visualization of interaction prediction results. The results in Fig. 4 show that 2G-GCN could even predicts entirely incorrect interactions across all the video frames (highlighted in read dashed boxes), while our model generates more precise predictions. Similarly, results in Fig. 5 show that our model avoids the unreasonable prediction results occur in 2G-GCN predictions.

## 4   SOURCE CODE OF OUR MODEL

We append our source code in the supplement materials.

## REFERENCES

Diederik P Kingma and Jimmy Ba. Adam: A method for stochastic optimization. *arXiv preprint arXiv:1412.6980*, 2014.

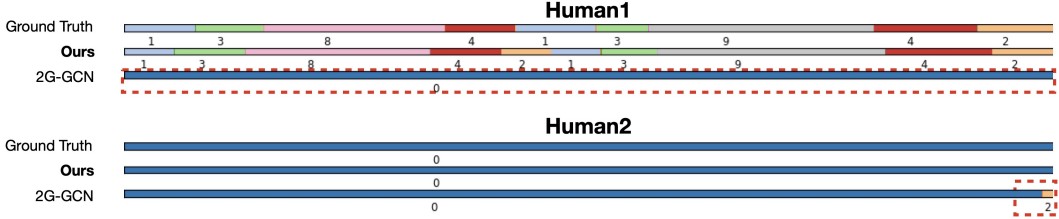

Figure 4: **Visualization results on MPHOI-72 dataset**. Major prediction errors are highlighted in red dashed boxes.

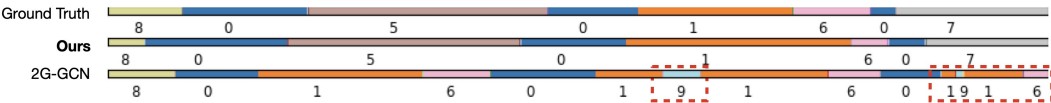

Figure 5: **Visualization results on CAD-120dataset**. Major prediction errors are highlighted in red dashed boxes.