# OpenReview forum: "Interaction-centric Hypersphere Reasoning for Multi-person Video HOI Recognition"
_ICLR.cc/2024/Conference — ICLR 2024 Conference Withdrawn Submission_

### Official Review · Reviewer_dGtW · 2023-10-26

**Soundness:** 2 fair
**Presentation:** 3 good
**Contribution:** 2 fair
**Rating:** 5
**Confidence:** 4

**Summary:**

This work focused on video HOI recognition and proposed a hypersphere-based method to learn the interdependency between humans, objects, and interactions. The authors proposed several modules like CF, ISR, and BiGRU to build a new pipeline to learn complex spatio-temporal video HOIs. On three benchmarks, the proposed method was evaluated and compared with previous works and showed improvements.

**Strengths:**

+ The complex relations within video HOIs are a meaningful problem for intelligent visual understanding, using hypersphere is an interesting attempt.

+ The whole paper is written well and easy to follow.

**Weaknesses:**

- Some design choices were not well illustrated and verified, which will be detailed in the questions.

- Some claims are ambiguous, please give more explanations:

usually lack of ability to capture global context information: which works and why?

ultimately compromising their representational accuracy: what is representational accuracy? Why Euclidean cannot?

**Questions:**

1. Why choose the hypersphere? Its pros upon Euclidean? Maybe discussions and experiments for support.

2. Each class has its own hypersphere, then how to embed the relations between classes, e.g., holding and grasping? Is the current setting reasonable?

3. How to handle the multi-label classifications given hyperspheres, and the long-tailed bias?

4. Discussion about the temporal action localization or segmentation? And some possible comparison between this line of works?

---

> ### Author Response · Authors · 2023-11-21
> **Thanks for the constructive comments. All the questions are addressed as follows.**
>
> * ***Capturing global context information.*** All of the CNN-based works listed  Section 2 "Related Works-HOI detection" lacks the ability to capture global context information due to the inherent local nature of the CNN structure.
>
> * ***Representational accuracy.*** Representational accuracy indicate the quality of learned representations of entities in the scence. Euclidean distance do not introduce structure bias of HOI into the model as hypersphere does (illustrated in the next question), ignoring valuable structure information of HOI for guiding predictions.
>
> * ***The reason for choosing hyperspher.***
>
>   **(1)** The interaction-centric hypersphere is crafted to introduce a **structure bias** of HOI into our model. Specifically, this bias asserts that **human-object pairs adhere to their respective interaction classes**. Consequently, the model is compelled to make predictions within the confines of this structure bias. It is essential to clarify that the hypersphere serves merely as a means of visualizing this bias. Alternately, various other techniques may be employed for visualizing the structure bias. Hence, our paper's fundamental proposition is to instill the desired structure bias, emphasizing that hypersphere is just one illustrative manifestation of this broader concept.
>
>   **(2)** The advantage of hypersphere over Euclidean distance is that the latter does not introduce such priors, ignoring valuable structure information of HOI for guiding predictions. In the revised paper, we conduct experiments of utilizing Euclidean distance by setting $\lambda=0$ in Equation (6). Results in Table 1, 2 and 3 (highlighed in red in the revised paper) show that utilizing Euclidean distance results in at least 7%, 10% and 3% $F_1$ score drop in MPHOI dataset, CAD-120 dataset and Bimanual Actions dataset, respectively. Thus, the effectiveness of the structure bias introduced by hypersphere have been validated. We also show the results below for better understanding:
>
>     |***MPHOI*** | $F_1 @10$  |  $F_1 @25$  |  $F_1 @50$  |
>     | ------------- | ------------- | --------------  | -------------- |
>     |**Ours**       | **91.6**$\pm$**0.9** |  **84.5**$\pm$**2.6** |  **67.7**$\pm$**2.2** |
>     |Ours ($\lambda=0$)  |  81.2$\pm$0.7  |   74.8$\pm$4.2  |   53.2$\pm$0.3  |
>
>     |***CAD-120*** | $F_1 @10$               |  $F_1 @25$  |  $F_1 @50$  |
>     | -------------    | -------------              | --------------  | -------------- |
>     |**Ours**          | **90.7**$\pm$**2.9** |  **88.1**$\pm$**2.8** |  **77.6**$\pm$**4.7** |
>     |Ours ($\lambda=0$)  |  72.0$\pm$4.4  |   65.0$\pm$6.9  |   48.6$\pm$6.3  |
>
>     |***Bimanual Actions*** | $F_1 @10$  |  $F_1 @25$  |  $F_1 @50$  |
>     | ------------- | ------------- | --------------  | -------------- |
>     |**Ours**       | **85.0**$\pm$**2.5** |  **82.9**$\pm$**2.9** |  **74.2**$\pm$**4.3** |
>     |Ours ($\lambda=0$)  |  76.7$\pm$5.2  |   74.3$\pm$6.0  |   65.2$\pm$6.3  |
>
> * ***Relations between classes.*** The relations between interaction classes are modeled by Interaction State Reasoner (ISR) module. ISR explicitly empowers the model to determine whether the current interaction should persist or transit to another interaction with a state interpolation module and a reasoner block shown in Figure 3.
>
> * ***Multi-label and long-tail.***
>
>      (1) At each time step, there is only one sub-activity for each person. Therefore there is no multi-label problem in HOI recognition task.
>
>      (2) We deal with long-tail problem with focal loss $\mathcal{L}_{cls}$, which is illustrated in Section 4.3.
>
> * ***Temporal action segmentation.*** Temporal action localization or segmentation works usually focus on a single action of interest happening in the video, which is a different task compared to video HOI recognition, where multiple actions could happen in the input video.

---

> > ### Comment · Reviewer_dGtW · 2023-11-22
> > **Post-rebuttal**
> >
> > Thanks for the responses. After reading the reviews and the responses from the authors, I tend to retain my initial rating.

---

> ### Author Response · Authors · 2023-11-23
>
> Thanks for your comments. I would be glad to answer any additional questions if you have.

---

### Official Review · Reviewer_NfZw · 2023-10-31

**Soundness:** 2 fair
**Presentation:** 2 fair
**Contribution:** 2 fair
**Rating:** 5
**Confidence:** 4

**Summary:**

This paper proposes an interaction-centric hypersphere reasoning model for multi-person video HOI recognition. The design of interaction-centric hypersphere explicitly directs the learning process towards comprehending the HOI manifold structures governed by interaction classes, a hitherto unexplored domain.

**Strengths:**

The method proposed in the paper is interesting.

**Weaknesses:**

The experimental evaluation is not comprehensive.
The presentation for some key concepts and ideas is unclear, which needs extensive improvement.

**Questions:**

Presentation:
1. The definition of the task is unclear. What is multi-person video HOI recognition? How to understand multi-person interaction and what is the specific expression? If it is multi-person interaction, what is the difference from the research direction of group action recognition? This work is anchored to explore multi-person interactive action recognition, so please clearly describe the task content and specific input and output.

In addition, regarding the definition of the task of this article, I have some idea until the fourth section. The previous sections do not elaborate on it, which is very good for understanding.

2. Why it is called a hypersphere? What is the meaning of hypersphere and does it have any theoretical implications? Hypergraphs and graphs are different theories. In this paper, what is the difference between the concept of a hypersphere and a sphere?

“The design of interaction-centric hypersphere explicitly directs the learning process towards comprehending the HOI manifold structures governed by interaction classes, a hitherto unexplored domain.” This hypersphere appears to be used to predict interaction probabilities. Please explain how it differs from traditional classifiers? This explanation is necessary since this hypersphere is the key idea. In addition, there is not much comparison, description, and argumentation between manifold structures and hypersphere theories in this paper. On the contrary, other modules explain more, which makes me wonder what is the core of the paper.

3. What are the HOI manifold structure, which has been mentioned several times in this paper? It is hard to understand. Is it related to Riemannian geometry? Please elaborate it. It is better to have a clear explanation.

4. ”To enhance the awareness of complex HOI structures in our representations, we introduce the Context Fuser (CF)...” Is there any connection between complex HOI and multi-person HOI?

5. “To facilitate interaction reasoning, we place the ISR module on top of the context fuser module, yielding entity representations capable of capturing interaction transition dynamics.” Does entity representation represent the characteristics of human and object entities? Or does it represent the transition characteristics of the same person or object between different states?
This sentence confuses me a lot about what exactly it represents.

6. “However, current video HOI recognition methods do not fully explore such inherent structural nature of HOI components. Instead, they often opt for disentangled representations for each component, which may have suboptimal representation capabilities.” It is recommended to visualize the problem to be solved, so that readers can understand it clearly.

7. In Figures 2 and 3, it is better to replace the letters with specific features. Using a large number of letters is too unintuitive and makes it difficult for readers to understand.

8. “We follow 2G-GCN to extract feature of humans and objects from backbone network”.
You used 2G-GCN to capture features, but the input {vt}t=1T seems to be a clip. Is the input of 2G-GCN a video? The output is the characteristics of people and objects in each frame of the video? Do ZH and ZO represent the characteristics of people and objects in each frame, or the characteristics of the entire video? I'm totally confused.

Experiments:
1. Although three datasets are compared, the algorithm is not fully verified. Why the VidHOI dataset is not used for comparison? This is a well-known video-based human-object interaction dataset.

2. There are no comparisons for this hypersphere module in the ablation experiments. It's the key component that needs comparative validation.

---

> ### Author Response · Authors · 2023-11-21
> **Thanks for the detailed comments. We answer all the questions as follows.**
>
> * ***Task definition.***
>
>     **(1)** As we mentioned in Section 4.1 ''Problem Formulation'' part in the paper, multi-person video HOI recognition means there are multiple persons in the scene and we aim to predict the temporal segmentation of each peron's sub-activity in that scene, indicating the person's interaction with the object.
>
>     **(2)** Note that we do not aim to predict *multi-person interaction*. Rather, we predict ***human-object inteaction*** represented by human sub-activity classes.
>
>    **(3)** It is important to underscore that multi-person HOI interaction stands apart from group action recognition [1][2]. Unlike the latter, which yields a singular action class for all individuals in the scene, multi-person HOI interaction goes a step further by prognosticating each person's distinct sub-activity in the given scene. This added granularity introduces a heightened level of complexity, surpassing the challenges inherent in group action recognition.
>
>    *[1] Azar, Sina Mokhtarzadeh, et al. "Convolutional relational machine for group activity recognition." Proceedings of the IEEE/CVF Conference on Computer Vision and Pattern Recognition. 2019.*
>
>    *[2] Gavrilyuk, Kirill, et al. "Actor-transformers for group activity recognition." Proceedings of the IEEE/CVF Conference on Computer Vision and Pattern Recognition. 2020.*
>
> * ***Hypersphere.***
>
>     **(1)** Hypersphere indicates a high-dimensional sphere, indicating the boundary of a high-dimensional ball. We call "hypersphere" in this work because the feature dimensions in Equation (6) are high-dimensional vectors (larger than 3).
>
>     **(2)** The interaction-centric hypersphere is crafted to introduce a **structured bias** of HOI into our model. Specifically, this bias asserts that **human-object pairs adhere to their respective interaction classes**. Consequently, the model is compelled to make predictions within the confines of this structured bias. It is essential to clarify that the hypersphere serves merely as a means of visualizing this bias. Alternately, various other techniques may be employed for visualizing the structure bias. Hence, our paper's fundamental proposition is to instill the desired structure bias, emphasizing that hypersphere is just one illustrative manifestation of this broader concept.
>
> * ***Differnece of hypersphere from traditional classifier.*** As we explained in the above section, hypersphere is a way to visualize the introduced structured bias underlying HOI. This structure bias forces the model to make predictions within the confines of this structured bias. Compared to tranditional classifiers which usually employ multi-layer perceptron (MLP) to generate prediction logits, hypersphere introduce a strong prior into model prediction, which is abscent in traditional classifiers. Moreover, hypersphere do not introduce any training parameters like traditional classifiers, featuring higher computational efficiency.
>
> * ***HOI manifold structure.*** In mathematics and machine learning, a "manifold structure" refers to a topological space that locally resembles Euclidean space. In other words, in the vicinity of any point on the manifold, the space behaves like a flat, n-dimensional Euclidean space. Manifolds can have various dimensions and shapes, and they are often used to represent complex data structures or patterns in high-dimensional spaces. In this work, *HOI manifold structure* specifically describes the topological relationship between **human-object paris** and **interaction class**, which is a structure bias that we introduce into our model.
>
> * ***Complex HOI.*** Complex HOI incorporates multi-person HOI, meaning there would be interdependencies between human and object in single-person cases, or interdependencies between human and human under multi-person scenarios.
>
> * ***Entities representations.*** Human and object are called "entity", so entity representations denote representation of human-object pair.

---

> ### Author Response · Authors · 2023-11-21
> **(Continued)**
>
> * ***Figure 2 and 3.*** We have added annotations  for the letters in the figures.
>
> * ***Follow 2G-GCN to extract feature.*** {${v_t}$}$_{t=1}^T$ denote video frames, whose features are extracted from 2G-GCN model. We use CLIP model to extract text features, not video frame features. The input of 2G-GCN is the same with our model, which is a video. The output is the human sub-activity class at each time step. $z_H$ and $z_O$ are extracted features of human and object from 2G-GCN, respectively.
>
> * ***Experiment on VidHOI dataset.*** VidHOI is used for video HOI detection task, while our paper works on video HOI recognition, which is a different task. Consequently, the VidHOI dataset is not suitable for comparison in our study.
>
> * ***Ablating hypersphere module.*** In the revised paper, we conduct an ablative analysis on the hypersphere module, substituting it with a conventional classifier implemented through a Multilayer Perceptron (MLP). The outcomes, as presented in Table 1, 2, and 3 (highlighted in red in the revised paper), reveal a notable decline of over 7% in the $F_1$ score within the MPHOI dataset when employing the traditional classifier. Likewise, in the CAD-120 dataset, the traditional classifier results in a substantial decrease of more than 11% in the $F_1$ score. Additionally, within the Bimanual Actions dataset, the traditional classifier induces a decline exceeding 2% in the $F_1$ score. These findings unanimously underscore the efficacy of the structural bias introduced by the hypersphere module. We also show the results below for better understanding:
>
>   |***MPHOI*** | $F_1 @10$  |  $F_1 @25$  |  $F_1 @50$  |
>   | ------------- | ------------- | --------------  | -------------- |
>   |**Ours**       | **91.6**$\pm$**0.9** |  **84.5**$\pm$**2.6** |  **67.7**$\pm$**2.2** |
>   |Ours (Traditional Classifier)  |  84.6$\pm$7.2  |   74.7$\pm$10.5  |   54.6$\pm$13.7  |
>
>   |***CAD-120*** | $F_1 @10$               |  $F_1 @25$  |  $F_1 @50$  |
>   | -------------    | -------------              | --------------  | -------------- |
>   |**Ours**          | **90.7**$\pm$**2.9** |  **88.1**$\pm$**2.8** |  **77.6**$\pm$**4.7** |
>   |Ours (Traditional Classifier)  |  79.5$\pm$11.0  |   73.9$\pm$11.4  |   56.6$\pm$12.5  |
>
>   |***Bimanual Actions*** | $F_1 @10$  |  $F_1 @25$  |  $F_1 @50$  |
>   | ------------- | ------------- | --------------  | -------------- |
>   |**Ours**       | **85.0**$\pm$**2.5** |  **82.9**$\pm$**2.9** |  **74.2**$\pm$**4.3** |
>   |Ours (Traditional Classifier)  |  82.0$\pm$3.6  |   79.8$\pm$4.1  |   71.0$\pm$5.6  |

---

> ### Author Response · Authors · 2023-11-23
>
> Thanks for your comments. I would be glad to answer any additional questions if you have.

---

### Official Review · Reviewer_dB2v · 2023-11-03

**Soundness:** 2 fair
**Presentation:** 2 fair
**Contribution:** 2 fair
**Rating:** 5
**Confidence:** 3

**Summary:**

This paper proposes an interaction-centric hypersphere reasoning model for multi-person video HOI recognition. To do this, a context fuser is designed to learn the interdependencies among humans, objects, and interactions; a state reasoner model on top of context fuser is used for temporal reasoning; an interaction-centric hypersphere is used to represent the manifold structure of HOIs.

The model is flexible for multi-person or single-person videos. Experiments show the method outperforms the previous method by 22% F1 score on multi-person dataset, MPHOI-72 and the method performs similarly with existing methods on single-person dataset, Bimanual Actions and CAD-120.

**Strengths:**

- The paper proposes an interaction-centric hypersphere representation scheme for HOI recognition learning.
- The method achieves SOTA performance with
a huge improvement of more than 22% F1 score over existing methods.

**Weaknesses:**

- The main focus of the paper is HOI recognition for multi-person videos. In the experiment, there is only 1 multi-person dataset used for evaluation but 2 single-person datasets. Showing model performance on different multi-person datasets will help strength the claims in the paper.
- After reading the paper, there is still a lack of proof or explanation about why an interaction-centric hypersphere will help in the task theoretically. The ablation study does not show an ablation study on it.

**Questions:**

- In the method, the context fuser and interaction state reasoner extract CLIP features for the representation. Ablation on the features is necessary to test if the feature is important or over-complex where a simple binary feature is enough. For example, in the interaction state reasoner, the two possible states “continue” and “stop” can be represented by binary labels or simpler features of lower dimensions compared with CLIP.
- In 4.3.1 Model inference, during model inference, the interaction probability is predicted on each frame. It is not clear who is in interaction if there are multiple people in the video. Interaction prediction for each person is more detailed and straightforward.
- Based on the question above, ablation studies with methods of HOI detection on images are necessary. HOI detection can detect interaction for each person. If there are multiple people, the results for comparison from the HOI detection is whether there is any interaction from all people or not.
- In the Conclusion Sec, it mentions that the method outperforms SOTA on the multiple-people dataset but is on par with the single-person dataset. Is it because in the single-person video, there is only one person so the interaction prediction is determined by that single person? But for multi-person videos, the model does not need to predict correctly for each person to get the correct answer for the image.

---

> ### Author Response · Authors · 2023-11-21
> **Thanks for the constructive comments. The questions are addressed in the following sections.**
>
> * ***Experiment datasets.*** In our multi-person video HOI recognition experiment, we exclusively leverage the MPHOI dataset, as it stands as the sole dataset tailored for multi-person video HOI recognition at present. Aligning with the SOTA methodology, specifically 2G-GCN, which also centers on multi-person video HOI recognition, our experimental design mirrors theirs by utilizing the MPHOI dataset. Furthermore, 2G-GCN augments its investigation with two single-person datasets. Hence, our experimental settings maintain congruence with 2G-GCN to facilitate a meaningful comparative analysis.
>
> * ***Why interaction-centric hypersphere helps?*** The interaction-centric hypersphere is crafted to introduce a **structure bias** of HOI into our model. Specifically, this bias asserts that **human-object pairs adhere to their respective interaction classes**. Consequently, the model is compelled to make predictions within the confines of this structured bias. It is essential to clarify that the hypersphere serves merely as a means of visualizing this bias. Alternately, various other techniques may be employed for visualizing the structure bias. Hence, our paper's fundamental proposition is to instill the desired structure bias, emphasizing that hypersphere is just one illustrative manifestation of this broader concept.
>
> * ***Ablating CLIP features.*** We conduct ablation studies in the revised peper by utilizing random initialization for interaction features and context features, instead of using CLIP representations. Ablation study results in Table 1 (highlighted in red in the revised paper) show that random initialization results in around 10% in the $F_1$ scores in MPHOI dataset, while still higher than SOTA method 2G-GCN around 15%. Furthermore, Tables 2 and 3 showcase ablation study outcomes on the CAD-120 and Bimanual Actions datasets, indicating a drop of less than 3% in $F_1$ scores with random initialization. Notably, these scores remain competitive, if not superior, to the 2G-GCN model. Thus, our findings suggest that extracting CLIP features is not imperative for achieving strong performance. We also show the results below for better understanding:
>
>   |***MPHOI*** | $F_1 @10$  |  $F_1 @25$  |  $F_1 @50$  |
>   | ------------- | ------------- | --------------  | -------------- |
>   | 2G-GCN     |  68.6$\pm$10.4 |  60.8$\pm$10.3 |  45.2$\pm$6.5 |
>   |**Ours**       | **91.6**$\pm$**0.9** |  **84.5**$\pm$**2.6** |  **67.7**$\pm$**2.2** |
>   |Ours ($\Delta$ CLIP+BLIP)  |  80.0$\pm$6.7  |   73.0$\pm$9.9  |   55.6$\pm$7.4  |
>
>   |***CAD-120*** | $F_1 @10$               |  $F_1 @25$  |  $F_1 @50$  |
>   | -------------    | -------------              | --------------  | -------------- |
>   | 2G-GCN     |  89.5$\pm$1.6 |  87.1$\pm$1.8 |  76.2$\pm$2.8 |
>   |**Ours**          | **90.7**$\pm$**2.9** |  **88.1**$\pm$**2.8** |  **77.6**$\pm$**4.7** |
>   |Ours ($\Delta$ CLIP+BLIP)  |  89.4$\pm$2.3  |   85.5$\pm$3.9  |   74.9$\pm$5.7  |
>
>   |***Bimanual Actions*** | $F_1 @10$  |  $F_1 @25$  |  $F_1 @50$  |
>   | ------------- | ------------- | --------------  | -------------- |
>   | 2G-GCN     |  **85.0**$\pm$**2.2** |  82.0$\pm$2.6 |  69.2$\pm$3.1 |
>   |**Ours**       | 85.0$\pm$2.5 |  **82.9**$\pm$**2.9** |  **74.2**$\pm$**4.3** |
>   |Ours ($\Delta$ CLIP+BLIP)  |  84.3$\pm$1.4  |   81.8$\pm$1.8  |   73.2$\pm$2.7  |
>
>
> * ***Interaction prediction for each person.***  In the case of multi-person video HOI recognition, our model is able to predict interaction class for each person, as qualitative examples shown in Figure 5 and Figure 7 in the paper.
>
> * ***Ablation studies with methods of HOI detecion.*** Our emphasis lies specifically on the **HOI recognition** task rather than HOI detection. These two tasks differs in both problem formulation and technical challenges. As articulated in Section 4.1 "Problem Formulation," our focus is on predicting the HOI class present in a given video frame. It is essential to clarify that our paper does not address HOI detection, which involves generating bounding box predictions for each individual and object in the scene, a distinct task from the scope of our work.
>
> * ***Performacne on single-person videos.*** The relatively modest improvement of our model's performance on single-person videos compared to the SOTA method in contrast to its notable gains in multi-person videos can likely be attributed to the absence of a dedicated context fuser module for single-person scenarios, as illustrated in Figure 2. As evident from the outcomes presented in Table 1, the contextual fusion module plays a pivotal role in enhancing performance in multi-person settings. Consequently, we hypothesize that the less remarkable results observed in single-person instances are attributable to the lack of a context fuser module.

---

> ### Author Response · Authors · 2023-11-23
>
> Thanks for your comments. I would be glad to answer any additional questions if you have.

---

### Official Review · Reviewer_3Vxe · 2023-11-05

**Soundness:** 1 poor
**Presentation:** 2 fair
**Contribution:** 2 fair
**Rating:** 3
**Confidence:** 4

**Summary:**

This paper presents a novel *Context Fuser* module leveraging the strengths of the CLIP and BLIP models, incorporates an *Interaction State Reasoner* module, and introduces *Interaction Feature Loss* to address the video Human-Object Interaction (HOI) problem.

**Strengths:**

The experimental results demonstrate superior performance over other methods on MPHOI-72 and CAD-120 benchmarks.

**Weaknesses:**

1. **Unfair Comparisons and Insufficient Ablation Studies:**
The primary weakness of the proposed method is the unfair comparisons with other video HOI methods and the lack of thorough ablation studies. The Context Fuser employs large Vision-Language Models (VLMs), CLIP and BLIP, pre-trained on big data. While ASSIGAN and 2G-GCN do not. In Table 1, removing Context Fuser results in a notable decline in $F_1@10$. This raises suspicions that the significant performance enhancement could be largely attributed to the text-image alignment capabilities inherent in large VLMs rather than the proposed Context Fuser. The performance is below the benchmark set by Qiao et al., 2020, in the absence of the Context Fuser. The paper lacks critical ablation studies to disentangle the contributions of the VLMs and the proposed method.
2. **Missing References:**
There is a relevant ICLR’23 paper you should refer to, Gao et al., ICLR’23. Gao et al, which also uses large VLMs for the HOI problem. Unlike fixed prompt used in Context Fuser, while Gao et al. delve into learnable prompts.
>Gao, K., Chen, L., Zhang, H., Xiao, J. & Sun, Q. Compositional Prompt Tuning with Motion Cues for Open-vocabulary Video Relation Detection. _ICLR_ (2023).

**Questions:**

A deeper ablation study focusing on the efficacy of the CLIP and BLIP parts within the Context Fuser is advisable. This would ascertain whether the observed improvements stem from the newly proposed module or merely from the integration of CLIP and BLIP.

---

> ### Author Response · Authors · 2023-11-21
> **Thanks for the constructive comments. We address the questions as follows.**
>
> * ***Comparison with other methods and ablation studies.***: In our revised paper, we employ ablation studies to elucidate the impact of excluding prominent Vision-Language Models (VLMs) CLIP and BLIP (highlighted in red in the revised paper). In the ablation studies, we adopt random initialization for the interaction feature $z_I$ and context feature $z_C$. The results, detailed in Table 1, indicate approximately 10% in $F_1$ scores within the MPHOI dataset upon removing CLIP and BLIP. Despite this reduction, our scores remain superior to the SOTA method 2G-GCN by approximately 15% $F_1$ score. Moreover, Tables 2 and 3 reveal that in the CAD-120 and Bimanual Actions datasets, the omission of CLIP and BLIP yields a drop of less than 3% in the $F_1$ scores, still comparable to or surpassing the performance of the 2G-GCN model. These findings underscore that the comparison between our model and SOTA methods is equitable, as the reliance on large VLMs proves non-essential for achieving commendable performance. Rather, it is the intricately designed structure of our model that substantiates the substantial enhancement in performance. We also show the results below for better understanding:
>
>   |***MPHOI*** | $F_1 @10$  |  $F_1 @25$  |  $F_1 @50$  |
>   | ------------- | ------------- | --------------  | -------------- |
>   | 2G-GCN     |  68.6$\pm$10.4 |  60.8$\pm$10.3 |  45.2$\pm$6.5 |
>   |**Ours**       | **91.6**$\pm$**0.9** |  **84.5**$\pm$**2.6** |  **67.7**$\pm$**2.2** |
>   |Ours ($\Delta$ CLIP+BLIP)  |  80.0$\pm$6.7  |   73.0$\pm$9.9  |   55.6$\pm$7.4  |
>
>   |***CAD-120*** | $F_1 @10$               |  $F_1 @25$  |  $F_1 @50$  |
>   | -------------    | -------------              | --------------  | -------------- |
>   | 2G-GCN     |  89.5$\pm$1.6 |  87.1$\pm$1.8 |  76.2$\pm$2.8 |
>   |**Ours**          | **90.7**$\pm$**2.9** |  **88.1**$\pm$**2.8** |  **77.6**$\pm$**4.7** |
>   |Ours ($\Delta$ CLIP+BLIP)  |  89.4$\pm$2.3  |   85.5$\pm$3.9  |   74.9$\pm$5.7  |
>
>   |***Bimanual Actions*** | $F_1 @10$  |  $F_1 @25$  |  $F_1 @50$  |
>   | ------------- | ------------- | --------------  | -------------- |
>   | 2G-GCN     |  **85.0**$\pm$**2.2** |  82.0$\pm$2.6 |  69.2$\pm$3.1 |
>   |**Ours**       | 85.0$\pm$2.5 |  **82.9**$\pm$**2.9** |  **74.2**$\pm$**4.3** |
>   |Ours ($\Delta$ CLIP+BLIP)  |  84.3$\pm$1.4  |   81.8$\pm$1.8  |   73.2$\pm$2.7  |
>
> * ***Reference.*** Thanks for the suggestion, we have added the reference in the revised version.

---

> ### Author Response · Authors · 2023-11-23
>
> Thanks for your comments. I would be glad to answer any additional questions if you have.

---

### Meta-Review · Area_Chair_qdZK · 2023-12-09

**Metareview:**

This paper received overall negative ratings (3, 5, 5, 5). The reviewers acknowledged that the paper tackles an interesting problem and that the proposed approach achieves strong performance. However, there was unanimous concern about the lack of detailed analyses and ablation studies justifying various design choices and important claims made in the paper. For example, why does the interaction-centric hypersphere help, and to what extent does it bring benefits (versus not using it)? How important is the context fuser versus the VLMs in contributing to the performance improvement?

The authors provided a rebuttal, showing additional ablation results and addressing some of the initial concerns. One reviewer maintained their initial rating after the rebuttal, while unfortunately, the three other reviewers did not update their ratings.

This meta-reviewer looked carefully at the reviews and the rebuttal and did not find it convincing enough to warrant acceptance. For instance, in response to reviewer `3Vxe`, the authors provided new results that do not use CLIP/BLIP and claimed that their approach is "still comparable to or surpassing the performance of the 2G-GCN model." However, upon inspecting results on the CAD-120 and Bimanual Actions, 2G-GCN outperforms the proposed approach except for one case (Bimanual Actions, F1@50). Similarly, in response to reviewer `NfZw`, the authors provide an ablation study that excludes the hypersphere module by replacing it with an MLP. Unfortunately, I could not find the details of the MLP to verify whether the comparison was fair.

After carefully reading the reviews and the rebuttal, this meta-reviewer did not find strong reasons to overturn the reviewers' recommendations.

**Justification For Why Not Higher Score:**

The rebuttal did not successfully address the reviewers' concerns.

**Justification For Why Not Lower Score:**

N/A

---

### Decision · Program_Chairs · 2024-01-16

Reject